# Palliative care need in the Eastern Mediterranean Region and human resource requirements for effective response

Eric L. Krakauer [1,2☯]*, Xiaoxiao J. Kwete [3☯], Maryam Rassouli [4☯], Héctor Arreola-Ornelas [5,6], Hadis Ashrafizadeh[7], Afsan Bhadelia[3], Yuwei A. Liu[8], Oscar Méndez-Carniado[5], Hibah Osman[1,9,10], Felicia M. Knaul [11]

1 Program in Global Palliative Care, Harvard Medical School, Boston, Massachusetts, United States of America, 2 Division of Palliative Care & Geriatric Medicine, Massachusetts General Hospital, Boston, Massachusetts, United States of America, 3 Harvard T.H. Chan School of Public Health, Boston, Massachusetts, United States of America, 4 School of Nursing & Midwifery, Shahid Beheshti University of Medical Sciences, Tehran, Iran, 5 Fundación Mexicana para la Salud, Mexico City, Mexico, 6 Institute for Obesity Research, Tecnologico de Monterrey, Mexico, Mexico, 7 School of Nursing, Dezful University of Medical Sciences, Dezful, Iran, 8 Yaozhi, Yangzhou, China, 9 Balsam Lebanese Center for Palliative Care, Beirut, Lebanon, 10 Dana Farber Cancer Institute, Boston, Massachusetts, United States of America, 11 Institute for Advanced Study of the Americas, University of Miami, Coral Gables, Florida, United States of America

☯ These authors contributed equally to this work.
* ekrakauer@hms.harvard.edu

**Data Availability Statement:** All original data are available to the public at the website of the International Association for Hospice & Palliative

## Abstract

Integration of palliative care into health care systems is considered an ethical responsibility, yet no country in the Eastern Mediterranean Region (EaMReg) has achieved integration. Data on palliative care need and cost are crucial forEaMReg health care planners and implementers in the region. Using data from the Lancet Commission on Palliative Care and Pain Relief, we estimated the number of people in each EaMReg country who needed palliative care in 2015 and their degree of access. In three countries, we estimated the number of days during which an encounter for palliative care was needed at each level of the health care system. This enabled calculation of the number of full-time equivalents (FTEs) of clinical and non-clinical staff members needed at each level to administer the essential package of palliative care recommended by WHO. In 2015, 3.2 million people in the EaMReg needed palliative care, yet most lacked access to it. The most common types of suffering were pain, fatigue, weakness, anxiety or worry, and depressed mood. To provide safe, effective palliative care at all levels of health care systems, between 5.4 and 11.1 FTEs of trained and supervised community health workers per 100,000 population would be needed in addition to 1.0–1.9 FTEs of doctors, 2.2–4.3 FTEs of nurses, and 1.4–2.9 FTEs of social workers. Data from our study enables design of palliative care services to meet the specific needs of each EaMReg country and to calculate the cost or cost savings.

Care: https://hospicecare.com/what-we-do/projects/global-data-platform-to-calculate-shs-and-palliative-care-need/database/.

**Funding:** The authors received no specific funding for this work.

**Competing interests:** The authors have declared that no competing interests exist.

## Introduction

Per the Amended Constitution of the World Health Organization (WHO), "health is a state of complete physical, mental and social well-being and not merely the absence of disease or infirmity" [1]. The opposite of health as wellbeing is therefore not just disease but suffering, and "enjoyment of the highest attainable standard of health," a fundamental right of every human being, requires prevention and relief of all types of suffering [1–3]. Accordingly, the World Health Assembly resolved unanimously in 2014 that palliative care (PC)–the prevention and relief of physical, psychological, social or spiritual suffering associated with serious illness–is "an ethical responsibility of health systems;" that "it is the ethical duty of health care professionals to alleviate pain and suffering, whether physical, psychosocial or spiritual, irrespective of whether the disease or condition can be cured;" that it is especially important to integrate PC into primary care and include home care; and that efforts to minimize risk of diversion of controlled medicines for illicit purposes must "not result in inappropriate regulatory barriers to medical access to such medicines" [4].

Based on these global agreement, ministries of health should base their work not just on the burden of disease but on the broader burden of suffering [5, 6]. Yet until recently, few estimates were available of the burden of suffering related to serious illness and the accessibility of palliative care to prevent and relieve it. The first edition of the WHO Global Atlas of Palliative Care estimated that 40 million needed palliative care in 2011, 78% of whom lived in LMICs, but that few of the 31 million in need in LMICs had access to it [7]. WHO's regularly repeated Noncommunicable Disease Country Capacity Survey also has demonstrated poor access to palliative care, especially in lower income settings [8]. This lack of access appears to have several causes: a lack of government policies recognizing palliative care as an essential service covered by health insurance and an official medical specialty; lack of required education in palliative care for medical and nursing students and practicing clinicians; overly restrictive laws and regulations governing accessibility of opioid analgesics; and a resultant lack of clinical services [7–10].

In the WHO Eastern Mediterranean Region (EaMReg), which consists of 22 member-states with widely differing cultures, health systems and resources, palliative care is inaccessible for most people in need [11]. The region includes high-income, petroleum-rich countries where palliative care is available in the larger cities but also low-income countries where political instability, conflict, and forced migration makes any health care difficult [11]. Overall, the EaMReg has far fewer palliative care services than the other five WHO regions. No country in the region has achieved integration of palliative care into its health care system, and only four countries have more than isolated palliative care provision [12]. Thus, there is a great need for data to guide integration of palliative care into health care systems in the region [4, 13].

Following the WHO Constitution, and to help fulfill the 2014 Resolution, the Lancet Commission on Palliative Care and Pain Relief (LC) undertook the first detailed estimates of the global burden of suffering associated with serious illnesses [5]. The method devised by the LC permits detailed estimation of the burden of suffering in every region and country [14, 15]. These data can enable ministries of health, care planners, and implementers to make the most efficient and effective use of their resources to maximize wellbeing, in part by designing and integrating into their health care systems palliative care services tailored to the specific needs of their population [3, 6]. But the data from the LC have not yet been analyzed for specific world regions.

In addition to estimating the global burden of suffering and thus the global need for palliative care, the LC also proposed an essential package of palliative care (EPPC) designed to safely prevent or relieve the most common and severe suffering, to be affordable and universally

accessible even in LMICs, and to provide financial risk protection for patients and families by enabling care in the community [5]. The EPPC consists of four interventions; a set of safe, effective, inexpensive, readily available medicines and equipment; social supports for patients and family caregivers living in extreme poverty; and the trained human resources necessary to effectively administer these. The EPPC has been endorsed by WHO and adapted for use in specific populations [3, 16–18]. The LC costed this package in countries of various income levels and found that it is affordable and feasible enough to make it universally accessible in any setting as a necessary step toward universal health coverage (UHC).

## Methods

We generated the first detailed estimates of the burden of suffering associated with serious illnesses and the resultant need for palliative care in each country in the EaMReg. We also estimated the human resources needed at each level of health care systems in each country to administer the basic package of palliative care. These estimates were based on mortality data from the EaMReg and used the methods developed by the LC, including literature review and consensus of a panel of ten experienced palliative care physicians from LMICs in Africa, Asia and the Americas [5, 14]. While basic overviews exist of the palliative care situation in the EaMReg, we know of no previous estimates of human resource requirements to enable universal access to palliative care for all in need [12, 19].

To estimate the global burden of suffering associated with serious illness and the global need for palliative care, the LC identified the 20 serious illness conditions listed in the International Classification of Diseases (ICD)-10 that most commonly result in moderate or severe physical, psychological, social, or spiritual suffering [5, 6, 20]. The 20 conditions identified by the LC accounted for 45 percent of global deaths in 2015. To determine the number of deaths per year from each condition, and hence gain insight into the burden of suffering and need for palliative care, the LC used mortality data from the WHO Global Health Estimates (GHE) for 2015 [21, 22], aligned these data with the ICD-10 conditions [23], and then estimated the types, prevalence, and duration of physical and psychological suffering resulting from each condition. This enabled estimation of the percentage of people who die from each condition (decedents) who need palliative care due to some type of physical or psychological suffering. The LC also identified the conditions that often lead to suffering even among non-decedents, defined as people who had the condition but did not die in 2015. The LC was not able to make similar estimates for social or spiritual suffering.

Based on the LC's estimates of the global burden of illness-associated suffering and need for palliative care, we estimated the burden of suffering and need for palliative care in the EaMReg and in each member state in the region except Palestine for which no data were available. Specifically, we estimated the number of decedents and non-decedents with each of the 20 conditions in each country who needed palliative care in 2015 and the percentage of palliative care need generated by each condition in each country. We also estimated the total number of days during which those in need of palliative care in each country experienced a physical or psychological symptom and the percentage of symptom-days in each country attributable to each symptom. We then calculated an upper and lower bound of the total number of days of suffering [5]. The lower bound was calculated based only on the symptom of longest duration associated with each condition ("at-least" symptom days). The upper bound was calculated by adding together the duration in days of all symptoms associated with each condition ("at-most" symptom days).

To help assess the current level of palliative care accessibility in the EaMReg, we used as a surrogate measure the *distributed-opioid oral morphine equivalents* (DOME), defined by the

International Narcotics Control Board as the quantity of strong opioids other than methadone that were imported by or manufactured in each country in 2015, with all opioids converted to the equivalent quantity of oral morphine using a standard equianalgesic dosing table [5]. DOME has been accepted as a surrogate measure of palliative care accessibility because strong opioids are considered the most essential of palliative medicines [3, 6, 12]. For each country, we calculated the DOME per patient in need of palliative care. We recognize that this is an inexact measure because not all patients in need of palliative care have pain or require an opioid and also because opioids are used for purposes other than palliative care such as intra-operative and post-operative analgesia.

Next, we calculated the amount of opioid in DOME that would be required to adequately treat all patients needing palliative care who had moderate or severe pain or refractory terminal dyspnea in each country. Using this as a standard, we calculated the percentage of need for opioid in DOME among patients needing palliative care that was met in each country. Finally, using the method developed by the LC, we estimated the number of days when an inpatient or outpatient palliative care encounter would be needed at each level of the health care systems of three countries to administer the EPPC effectively to all patients in need. We chose one low-income country (Afghanistan), one lower-middle-income country (Morocco), and one upper-middle-income country (Jordan). These estimates enabled us to estimate the full-time-equivalents (FTEs) of doctors, nurses, social workers, psychologists, spiritual supporters, physical therapists, pharmacists, community health workers, clinical support staff, and non-clinical staff needed to effectively administer the EPPC to all patients in need. By estimating the number of inpatient days and outpatient visits required by patients needing palliative care with each illness condition at each location of care (referral hospital, provincial hospital, district hospital, community health centers and in the home), we also were able to estimate the palliative care work load for each staff category required at each of those locations, expressed in FTEs [13, 14]. In addition, we calculated the FTEs of each type of staff member per 100,000 population for the three countries.

## Results

In the EaMReg in 2015, over 1.6 million decedents and almost 1.6 million non-decedents needed palliative care, a total of 3.2 million (**Table 1**). In general, the number of persons in need of palliative care in a country reflected the population size. For example, Pakistan's total population in 2015 (approximately 189 million) and number of people in need of palliative (approximately 869,000) were both around twice as large as those of Egypt, the second most populated country in the region. Similarly, the countries with the Region's lowest population, Bahrain (approximately 1.4 million) and Qatar (approximately 2.4 million), also had the lowest and second lowest number of people in need of palliative care, respectively. However, countries where political violence was occurring, such as South Sudan, Syria and Yemen, had greater palliative care needs than would be expected from population alone. The conditions that most commonly generated suffering requiring palliation were malignant neoplasms, injuries, dementia, and cardiovascular disease (**Fig 1**). However, HIV/AIDS was the most common cause of palliative care need in a few countries including South Sudan where it generated over 70% of the need.

The LC identified 11 types of physical suffering and 4 psychological types that are the most common reasons for palliative care need (**Fig 2**) [5]. The LC estimated that 80 percent of patients who die of malignant neoplasms suffer from moderate or severe pain for an average of three months. The LC also concluded that moderate or severe pain is common among those who die from HIV/AIDS, injuries, disease of the liver, central nervous system, and

**Table 1. Numbers of symptom days and of decedents and non-decedents in need of PC in each country.**

| Countries | | At most symptom days (millions) | At least symptom days (millions) | Decedents (000) | Non- Decedents (000) | Total (Decedent and non-decedent) (000) |
|---|---|---|---|---|---|---|
| Low Income Countries | Afghanistan | 40 | 13 | 95 | 71 | **166** |
| | Palestine | No data | No data | No data | No data | **No data** |
| | Somalia | 24 | 8 | 47 | 51 | **98** |
| | South Sudan | 78 | 23 | 51 | 201 | **251** |
| Lower-Middle Income Groups | Egypt | 142 | 43 | 254 | 174 | **428** |
| | Morocco | 55 | 17 | 77 | 87 | **164** |
| | Pakistan | 248 | 75 | 513 | 355 | **869** |
| | Syrian Arab Republic | 26 | 8 | 51 | 72 | **123** |
| | Sudan | 60 | 19 | 109 | 120 | **228** |
| | Tunisia | 21 | 7 | 29 | 34 | **63** |
| | Yemen | 26 | 9 | 56 | 49 | **109** |
| Upper-Middle Income Countries | Iran (Islamic Republic of) | 125 | 38 | 155 | 208 | **363** |
| | Iraq | 33 | 11 | 71 | 72 | **143** |
| | Lebanon | 11 | 3 | 14 | 17 | **31** |
| | Jordan | 8 | 2 | 12 | 12 | **24** |
| | Libya | 8 | 3 | 13 | 13 | **26** |
| High Income Countries | Bahrain | 1 | 0 | 1 | 1 | **2** |
| | Kuwait | 2 | 1 | 3 | 4 | **7** |
| | Oman | 3 | 1 | 4 | 5 | **9** |
| | Qatar | 1 | 0 | 1 | 2 | **3** |
| | Saudi Arabia | 24 | 8 | 40 | 44 | **84** |
| | United Arab Emirates | 3 | 1 | 6 | 5 | **11** |
| Total | | 940 | 289 | 1602 | 1596 | **3198** |

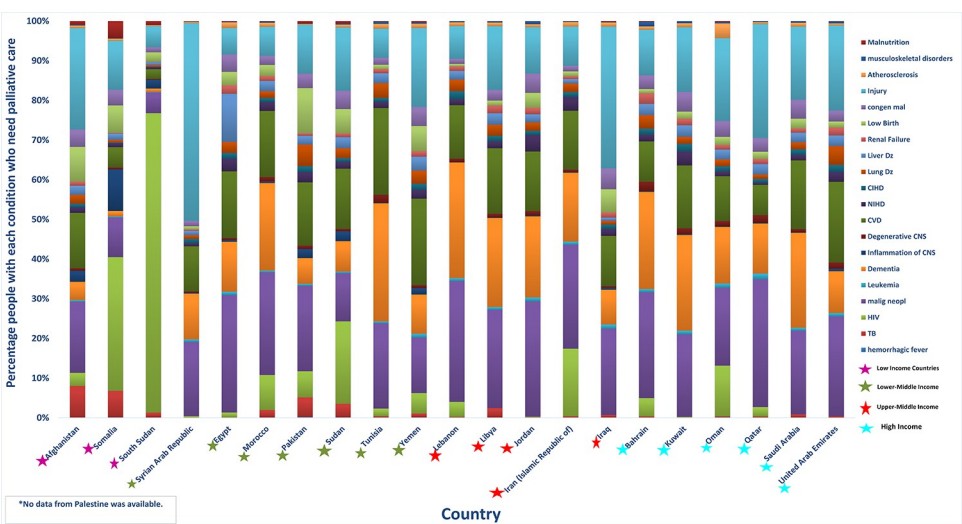

**Fig 1. Percentage of people (decedents & non-decedents) with each illness who needed palliative care in 2015, by country.** TB (Tuberculosis), CNS (Central nervous system), CVD (Cardiovascular diseases), NIHD (Non-ischemic heart disease), CIHD (chronic-ischemic heart disease).

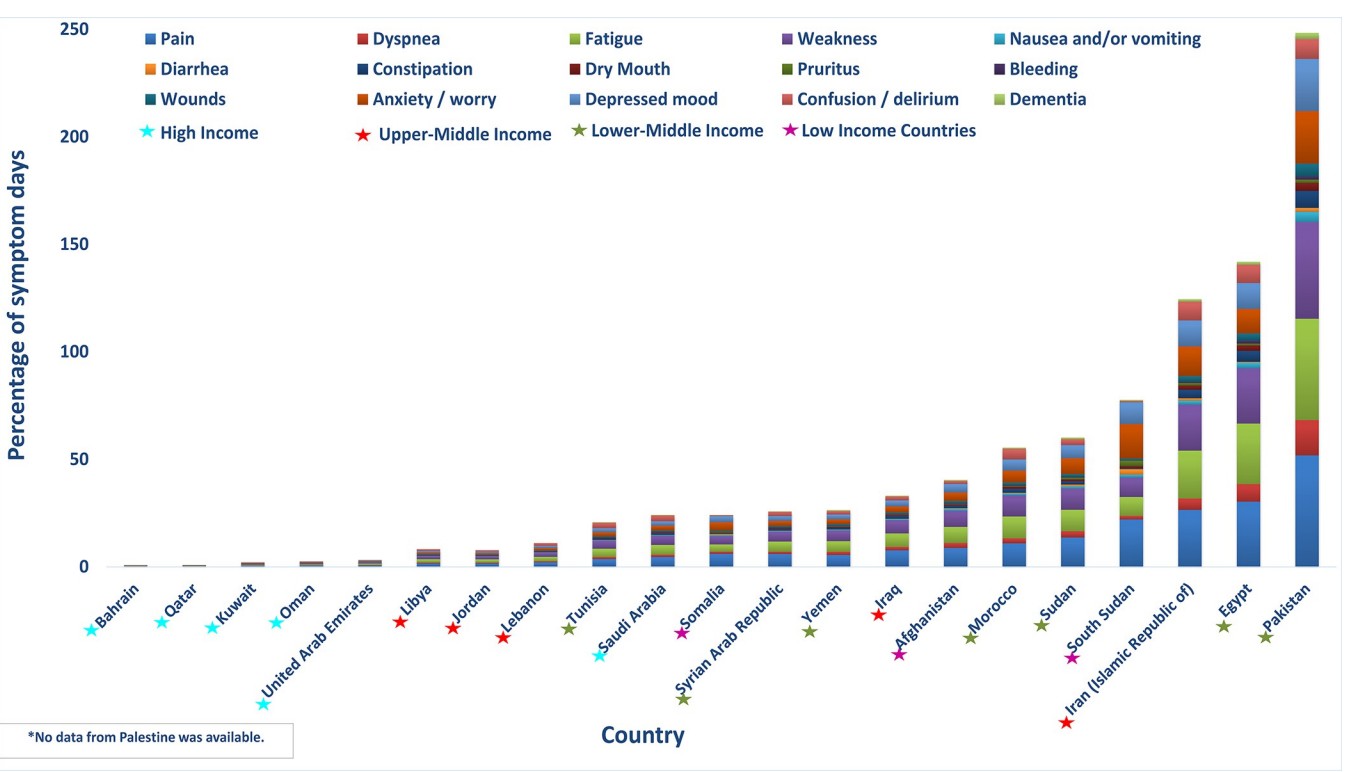

**Fig 2. Percentage of days with each type of physical and psychological suffering, by each country.**

musculoskeletal systems, congenital anomalies, and prematurity or birth trauma. Moderate or severe dyspnea—shortness of breath—was determined to be especially common among people who die of lung disease (100%), heart disease (75–80%) and only slightly less common among those who die of HIV/AIDS (70%), liver disease (50%), and prematurity or birth trauma (50%). Depressed mood and anxiety also were determined to be common among patients who died from malignant neoplasms, HIV/AIDS, dementia, other central nervous system disease, chronic lung disease, and other serious illnesses [5].

Our analysis showed that the most prevalent types of moderate or severe physical or psychological suffering that generated a need for palliative care in the EaMReg were pain, fatigue, weakness, anxiety or worry, and depressed mood (**Fig 2**). There were more days with pain than with any other physical or psychological symptom. Pain combined with fatigue and weakness accounted for more than half of the symptom days. However, anxiety/worry combine with depressed mood accounted for approximately one quarter of symptom days.

The distributed-opioid oral morphine equivalents (DOME) per patient in need of palliative care were inadequate to meet the estimated need in most countries in the EaMReg, including all low-income countries and lower-middle-income countries and all but one upper-middle-income country (Jordan) (**Fig 3**). With the exception of Oman all high-income countries had more than 100% of the DOME per patient in need of palliative care. However, DOME does not take into account that opioids are used for purposes other than palliative care, such as intra-operative and post-operative analgesia and thus that DOME is an artificially low measure of the total amount of medicinal opioid potentially available per patient receiving palliative care.In Afghanistan, a low-income country, 172,541 patients needed palliative care in 2015, 93,373 decedents and 77,439 non-decedents (**Table 2**). Effective administration of the EPPC to

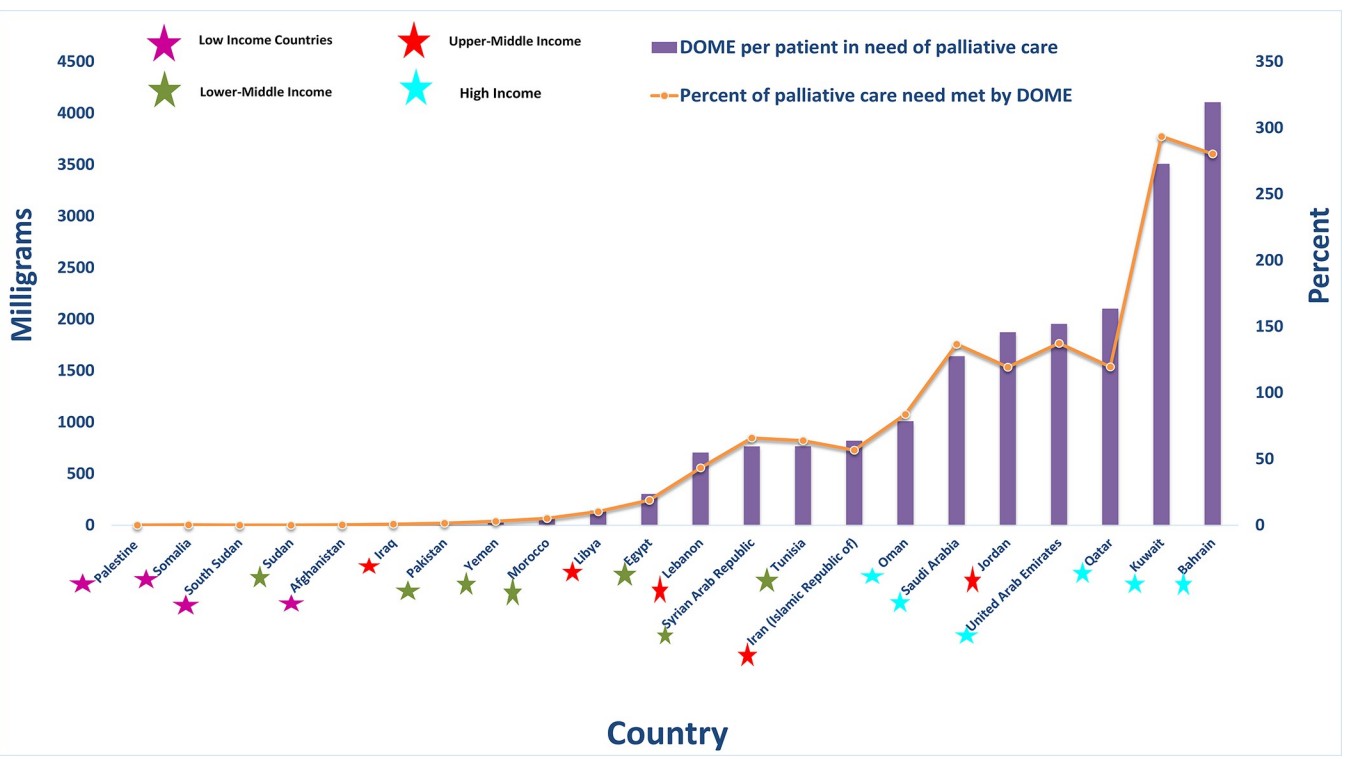

**Fig 3. Total amount DOME per patient in need of PC and percent SHS need met by DOME (%) for each country.**

everyone in need would have required an estimated 818,291 inpatient days with a palliative care encounter at referral hospitals and 194,418 palliative care outpatient visits at referral hospitals. A total of 5,431,684 home visits for palliative care would have been necessary. A total of 722 full-time equivalents (FTEs) of doctors with at least basic training in palliative care, distributed across the four levels of the health care system, would have been needed to respond to the entire need. Twice as many FTEs of nurses would have been needed, and 3018 FTEs of community health workers (CHWs). Per 100,000 population, 1.85 doctor FTEs, 3.71 nurse FTEs, 2.42 social worker FTEs, and 7.75 community health workers FTEs would have been needed. Table 2 also shows the estimates for one lower-middle-income country (Morocco) and one upper-middle-income country (Jordan).

## Discussion

There is a vast unmet need for palliative care to prevent and relieve moderate and severe suffering that is highly prevalent among people with serious illnesses in the EaMReg and around the world. A gradient in availability of palliative care has been shown, from high-income countries with the greatest availability to low-income countries with the least availability [12]. In addition to the EaMReg, three other WHO regions have very low availability of palliative care services and providers per population: the African, Southeast Asian, and Western Pacific Regions [12]. In the EaMReg, the unmet needs are greatest in LMICs and especially in countries where political violence is occurring. The 20 serious illnesses that result in most of the need for palliative care commonly cause multiple types of physical, psychological, social and spiritual suffering. Pain is the most common type of suffering and is especially common among people with cancer, injuries, and HIV/AIDS and among children with congenital

**Table 2. Decedents, nondecedents, and total patients needing PC in selected countries of different income levels; estimated number of days when a PC encounter would be needed at different sites; and human resources needed to provide PC.**

| Population in Need of Palliative Care | | Region | | |
|---|---|---|---|---|
| | | Afghanistan (LIC) | Morocco (LMIC) | Jordan (UMIC) |
| Total population | | 38,928,340 | 36,910,560 | 10,203,140 |
| Decedent | | 93373 | 76109 | 12283 |
| Non-Decedent | | 77439 | 87346 | 11753 |
| Total number of patients that need pc | | 172541 | 163898 | 22339 |
| Referral Hospital | Total inpatient days | 818291 | 586533 | 95847 |
| | Total outpatient visit | 194418 | 264207 | 33539 |
| Provincial Hospital | Total inpatient days | 567408 | 302913 | 48218 |
| | Total outpatient visit | 164497 | 215410 | 24623 |
| District Hospital | Total inpatient days | 502872 | 599917 | 86206 |
| | Total outpatient visit | 353407 | 473602 | 60769 |
| Community Health Center | Total inpatient days | 11633 | 14801 | 2229 |
| | Total outpatient visit | 375581 | 566372 | 68241 |
| Total home visits | | 5431684 | 5431684 | 997603 |
| FTE of staff involved in providing PC | Doctors—Specialists | 722 | 682 | 100 |
| | Nurses | 1443 | 1588 | 222 |
| | Social Workers | 940 | 1051 | 147 |
| | Spiritual Counsellor | 112 | 91 | 14 |
| | Psychologist or psychiatrist | 77 | 49 | 8 |
| | Physical Therapist | 27 | 17 | 3 |
| | Pharmacist | 168 | 164 | 24 |
| | Community Health Workers | 3018 | 4096 | 554 |
| | Clinical Support Staff (diagnostic imaging, Lab) | 12 | 10 | 2 |
| | Non Clinical Support Staff (Housekeeping, administration, Dietary) | 142 | 119 | 18 |
| Number of FTE per 100,000 population involved in providing PC | Doctors—Specialists | 1.85 | 1.85 | 0.98 |
| | Nurses | 3.71 | 4.30 | 2.18 |
| | Social Workers | 2.42 | 2.85 | 1.44 |
| | Spiritual Counsellor | 0.29 | 0.25 | 0.14 |
| | Psychologist or psychiatrist | 0.20 | 0.13 | 0.08 |
| | Physical Therapist | 0.07 | 0.04 | 0.03 |
| | Pharmacist | 0.43 | 0.44 | 0.23 |
| | Community Health Workers | 7.75 | 11.10 | 5.43 |
| | Clinical Support Staff (diagnostic imaging, Lab) | 0.03 | 0.03 | 0.02 |
| | Non Clinical Support Staff (Housekeeping, administration, Dietary) | 0.36 | 0.32 | 0.18 |

anomalies and prematurity or birth trauma. Anxiety and depression also are very prevalent among people with the 20 conditions. Dyspnea is not one of the most common types of suffering in general, but it is highly prevalent among people who die of some illnesses including lung disease, heart disease, HIV/AIDS, liver disease, and prematurity or birth trauma. Clinicians at all levels of health care systems should have the training and means to prevent and relieve all of these types of suffering [4, 13, 24].

Based on work by the LC, the World Bank's Disease Control Priorities and WHO recommend an EPPC that includes essential interventions, medicines, equipment, social supports, and trained human resources that should be universally accessible [3, 5, 6]. The most

important of the essential palliative medicines are oral fast-release and injectable morphine, a strong opioid. Using data from the International Narcotics Control Board to calculate the distributed-opioid oral morphine equivalents (DOME) per patient in need of palliative care, we found that access to strong opioids was inadequate to meet the estimated need in most countries in the EaMReg. If the amount of opioid used for invasive procedures could be subtracted from the DOME per patient in need of palliative care to make it a more accurate measure of the opioid required for palliative care only, even fewer countries might meet the estimated need.

The EPPC was designed to help enable health care planners, implementers and managers in the EaMReg to integrate effective palliative care into their health care systems and make it universally accessible. To further assist with this crucial task, we estimated the prevalence and duration of each type of physical and psychological suffering associated with each of the 20 conditions that most commonly generate a need for palliative care. Using these data, and by determining the cost to the health care system of the medicine and equipment in the EPPC, planners and implementers can calculate the cost to make these medicines and equipment universally accessible. In addition, we estimated the FTEs of all clinical and non-clinical staff members at all levels of health care systems needed to administer the EPPC in countries of three different income categories. Thus, managers and implementers in the EaMReg will be able to estimate the entire cost, or cost-savings, of making the EPPC universally accessible [5, 25, 26]. To our knowledge, our estimates of human resource needs to make basic palliative care universally accessible within each country in a region, including LMICs, are the first such estimates. Existing literature describes only palliative care specialist needs, or needs in special circumstances, in a few high-income countries [27–29].

Following WHO guidance, the EPPC can and should be provided not only in hospitals but also at community health centers and in the home if a patient's suffering can be adequately controlled in the community and the family would not be physically or psychologically overwhelmed caring for the patient at home [3, 4, 13, 24]. The crucial importance of community health workers (CHWs) to enable safe home care is reflected in the high number of estimated FTEs for CHWs (3018 CHW FTEs in Afghanistan, 4096 in Morocco, 554 in Jordan). Community health workers, who can visit patients at home as often as daily, can be trained to recognize and report to a supervisor at the community health center (CHC) any uncontrolled suffering, including social suffering such as severe poverty or social isolation, or improper use of medicines. In this way, they can serve as the eyes and ears of clinicians at the CHC [4, 24]. Home visits also provide an opportunity for CHWs to provide family members with illness prevention such as HPV vaccination, with screening for diseases such as cervical cancer and tuberculosis by collecting specimens, and with assistance to avoid missing treatment appointments. In these ways, among others, palliative care can and should be integrated with disease prevention, early diagnosis and treatment [2, 17, 18, 24].

Our study has several limitations. The data on which we based our estimates of the burden of suffering and need for palliative care already are several years old. However, we doubt that the situation has changed significantly over this period. We lacked any data from Palestine, and we were not able to estimate the prevalence or duration of moderate or severe social or spiritual suffering. Thus, our data set was incomplete. However, the EPPC is designed to include social supports to address social suffering and spiritual support, and the human resources to provide these supports are included in our estimates. The LC provides data on the specific cost of social supports in LMICs [5]. We acknowledge that the DOME per patient in need of palliative care is not a precise measure of palliative care development or quality in a country for many reasons, and we welcome work to improve these measurements.

## Conclusion

There is a large burden of suffering in the EaMReg associated with serious illnesses. Much of this suffering could be prevented or relieved with palliative care that currently is unavailable to most in the region. Thus, it is medical and morally imperative to make palliative care universally accessible as part of universal health coverage in the EaMReg. Data from our study and from the LC can enable health care planners, implementers and managers to design palliative care services to meet the specific needs of their population, to integrate these services into the existing health care system at all levels, and to calculate the cost or cost savings.

## Acknowledgments

We are grateful to Dr. Nasim Pourghazian, Dr. Lamia Mahmoud, and Dr. Slim Slama of the World Health Organization (WHO) Eastern Mediterranean Regional Office (EMRO) for their encouragement for this study. All views expressed are those of the authors alone and do not necessarily reflect those of WHO.

## Author Contributions

**Conceptualization:** Eric L. Krakauer, Xiaoxiao J. Kwete, Maryam Rassouli, Yuwei A. Liu, Hibah Osman, Felicia M. Knaul.

**Data curation:** Xiaoxiao J. Kwete, Maryam Rassouli, Héctor Arreola-Ornelas, Hadis Ashrafizadeh, Afsan Bhadelia, Yuwei A. Liu, Oscar Méndez-Carniado.

**Formal analysis:** Xiaoxiao J. Kwete, Maryam Rassouli, Héctor Arreola-Ornelas, Hadis Ashrafizadeh, Afsan Bhadelia, Yuwei A. Liu, Oscar Méndez-Carniado.

**Investigation:** Eric L. Krakauer, Afsan Bhadelia, Yuwei A. Liu, Hibah Osman, Felicia M. Knaul.

**Methodology:** Eric L. Krakauer, Xiaoxiao J. Kwete, Maryam Rassouli, Héctor Arreola-Ornelas, Hadis Ashrafizadeh, Yuwei A. Liu, Oscar Méndez-Carniado.

**Project administration:** Eric L. Krakauer, Felicia M. Knaul.

**Supervision:** Eric L. Krakauer, Felicia M. Knaul.

**Writing – original draft:** Eric L. Krakauer, Xiaoxiao J. Kwete, Maryam Rassouli, Héctor Arreola-Ornelas, Hadis Ashrafizadeh, Afsan Bhadelia, Yuwei A. Liu, Oscar Méndez-Carniado, Hibah Osman, Felicia M. Knaul.

**Writing – review & editing:** Eric L. Krakauer, Xiaoxiao J. Kwete, Maryam Rassouli, Hibah Osman, Felicia M. Knaul.

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
