## [Decision Letter · Decision Letter 0]

31 Jul 2023

PGPH-D-23-00751

Palliative Care Need in the Eastern Mediterranean Region and Human Resource Requirements for Effective Response

Dear Dr. Krakauer,

Thank you for submitting your manuscript to PLOS Global Public Health. After careful consideration, we feel that it has merit but does not fully meet PLOS Global Public Health’s publication criteria as it currently stands. Therefore, we invite you to submit a revised version of the manuscript that addresses the points raised during the review process.

We look forward to receiving your revised manuscript.

Kind regards,

Manish Barman, MD., MSc., FRCP

Academic Editor

Journal Requirements:

2. Please send a completed 'Competing Interests' statement, including any COIs declared by your co-authors. If you have no competing interests to declare, please state "The authors have declared that no competing interests exist". Otherwise please declare all competing interests beginning with twhe statement "I have read the journal's policy and the authors of this manuscript have the following competing interests:"

3. In the online submission form, you indicated that "All original data, including de-identified responses from individual experts, are available upon reasonable request to the corresponding author". All PLOS journals now require all data underlying the findings described in their manuscript to be freely available to other researchers, either 1. In a public repository, 2. Within the manuscript itself, or 3. Uploaded as supplementary information.

Additional Editor Comments (if provided):

Dear Authors

The paper is well-written on a topic which is essentially a part of personalized medicine and encompasses WHO global strategy on people-centered and integrated health services.

Your manuscript has undergone peer review process and I agree with our esteemed reviewers' suggestions.

Kindly address the concerned raised by the reviewers and we will be happy to consider it again at our earliest.

Thanks

Manish Barman

Reviewers' comments:

Reviewer's Responses to Questions

**Comments to the Author**

1. Does this manuscript meet PLOS Global Public Health’s publication criteria? Is the manuscript technically sound, and do the data support the conclusions? The manuscript must describe methodologically and ethically rigorous research with conclusions that are appropriately drawn based on the data presented.

Reviewer #1: Yes

Reviewer #2: Partly

Reviewer #3: Yes

2. Has the statistical analysis been performed appropriately and rigorously?

Reviewer #1: Yes

Reviewer #2: Yes

Reviewer #3: Yes

3. Have the authors made all data underlying the findings in their manuscript fully available (please refer to the Data Availability Statement at the start of the manuscript PDF file)?

Reviewer #1: Yes

Reviewer #2: No

Reviewer #3: Yes

4. Is the manuscript presented in an intelligible fashion and written in standard English?

Reviewer #1: Yes

Reviewer #2: Yes

Reviewer #3: Yes

5. Review Comments to the Author

Reviewer #1: Line 84- 86

The author should expand/ explain the challenges with access to Palliative care in the EMR region and the other five WHO regions. The literature review should provide more context to some of the drivers of poor access.

Aslo define how access to palliative care is defined globally. This will be relevant to comparing against the measure described in the the methodology section line 123 and 124

Line 158 – Table 1

It might be helpful to reflect the population size of each of the countries to provide a sense of prevalence of need

Line 212 Discussion section

The author could expand on how these findings relate with other publications on the same across other WHO regions

Reviewer #2: Authors need to work on solid rationale. The concept sounds but rationale to start the study is weak. Some other statistiscal tool can be used instead of presently used things.

Result and discussion sections need to more improvements and need technically sound.

It should be more good if authors prefer to compare UIC rather than lower and middle

Future implications need some more things which will be auspicious for upcoming authors.

Reviewer #3: Dear authors,

Subject: Palliative Care Need in the Eastern Mediterranean Region and Human Resource Requirements for Effective Response

Thank you for the opportunity to review this manuscript regarding this study on the palliative care need and staffing requirements in the Eastern Mediterranean Region (EMR). Please see the comments for suggested improvements to the manuscript:

Major Comments:

Overall, this paper was well-written, and well anchored in the literature. The methods section could benefit from referencing how staffing estimates were calculated (eg. on pg 7, line 144-145, it would be helpful to reference a formula, framework, or instrument used to find the number of FTE required). This would greatly reduce reader confusion on how those values were attained.

Comments by Section:

Introduction:

The description around Palliative Care within the Eastern Mediterranean Region was sufficient within the Introduction.

The context behind why the EMR is the focus of this paper is brought at the very end of the Introduction. It may be helpful to mention the importance of the palliative care needed within this region earlier in the Introduction and further elaborating on the challenges currently in place.

Within this region, what percentage of the population is within the EMR system and how often is EMR typically used? Answering this question within the Introduction would allow readers to better understand how frequently the EMR is being utilized and also shed light on the percentage of patient population that may not be frequently receiving healthcare.

The last few sentences in the Introduction (lines 89-92) may be better suited for the Methods section as they describe what is done in this study.

Methods:

Some concepts within the Methods could be better explained (such as the distributed-opioid oral morphine equivalents (DOME)) to give the reader more context to better understand how the study was conducted.

Please be more explicit about the data sources used. It is not clear enough from a reproducibility stand point.

Results:

Table 1 and Figure 2 show Pakistan to almost appear as an outlier in the upper and lower bound of symptom days, yet there isn’t really any discussion of trends in this data. If possible, it may be helpful to briefly mention or provide some context for such observations in the results section to improve reader understanding and allow a more comprehensive overview of EMR palliative needs.

If there is any numerical data on the current state of palliative care in the EMR (eg. current reported number of FTE PC staff in one of the listed countries), it might help to provide this information to compare current staffing with required staffing. This might make the EMR palliative care need/requirements more apparent to readers.

I liked the consideration of the FTEs within this paper.

Discussion:

Although this paper is focused on the palliative care needs within the EMR, it would be interesting to draw parallels and insights from other regions that have palliative care needs and describe the current research and interventions attempting to better integrate palliative care in those regions. This may help better contextualize this study within ongoing work in this field in other regions around the world.

Did the EMR report all patient data, or would some patient data not be included within the EMR? If not all patients are included, it would be worthwhile to include this within the limitations paragraph of the discussion.

Grammar, Spelling, Punctuation Edits:

Pg 10, line 202 “A total of 772 full-time…”. It says “772” FTE doctors needed, but Table 2 below says “722” FTE doctors for doctor-specialists under the column “Afghanistan-LIC”. This seems like a typo (772 ≠ 722) that can be corrected.

Table 2 - Just for consistency, it might be helpful to capitalize “pc” on the left side of the table

6. PLOS authors have the option to publish the peer review history of their article (what does this mean?). If published, this will include your full peer review and any attached files.

**Do you want your identity to be public for this peer review?** For information about this choice, including consent withdrawal, please see our Privacy Policy.

Reviewer #1: No

Reviewer #2: No

Reviewer #3: No

---

## [Editor Report · Decision Letter 1]

12 Oct 2023

Palliative Care Need in the Eastern Mediterranean Region and Human Resource Requirements for Effective Response

PGPH-D-23-00751R1

Dear Dr. Krakauer,

We are pleased to inform you that your manuscript 'Palliative Care Need in the Eastern Mediterranean Region and Human Resource Requirements for Effective Response' has been provisionally accepted for publication in PLOS Global Public Health.

Best regards,

Manish Barman, MD., MSc., FRCP

Academic Editor

Dear Authors

Thank you for methodically addressing the concerns raised by the esteemed reviewers and subsequent corrections.

Best

Manish Barman